# Intimate partner violence (IPV) with miscarriages, stillbirths and abortions: Identifying vulnerable households for women in Bangladesh

Awan Afiaz[1]*, Raaj Kishore Biswas[2], Raisa Shamma[1], Nurjahan Ananna[3]

1 Institute of Statistical Research and Training, University of Dhaka, Dhaka, Bangladesh, 2 Transport and Road Safety (TARS) Research Centre, School of Aviation, University of New South Wales, NSW, Australia, 3 Ibrahim Medical College, Dhaka, Bangladesh

☯ These authors contributed equally to this work.
* aafiaz@isrt.ac.bd

**Data Availability Statement:** The secondary data sets analyzed during the current study are freely available upon request from the DHS website at https://dhsprogram.com/data/available-datasets.

## Abstract

Intimate partner violence (IPV) is a social problem in Bangladesh with adverse effects on maternal healthcare. This study analyzed the sociodemographic factors responsible for intimate partner violence and its overall association with reproductive healthcare–specifically miscarriages, stillbirths and induced abortions (MSA)–using Bangladesh Demographic Health Survey 2007, which contains the latest available intimate partner violence data till date, with the hypothesis that intimate partner violence is associated with miscarriages, stillbirths and induced abortions. The generalized linear regression model was fitted to 3,920 women adjusting survey weights and cluster/strata variations. The study concluded that 1 out of every 4 women who reported experiencing intimate partner violence also reported having one or more of miscarriages, stillbirths and induced abortions. The results revealed that intimate partner violence and miscarriages, stillbirths and induced abortions were significantly associated with the age of the women, residence, age of the women at their first birth, sex of household head and the household's financial condition. Furthermore, the odds of having one or more miscarriages, stillbirths and abortions was increased by 35% for women who were victims to intimate partner violence, establishing a significant association between miscarriages, stillbirths and abortions and intimate partner violence. There appeared to be a need to address the issue in both paradigms, particularly for the poor rural women in Bangladeshi patriarchal society. These findings demand a combined intervention effort in the vulnerable cohorts, especially if Bangladesh intends to attain the goals 3.1 and 5.2 of the Sustainable Development Goals (SDG) by 2030.

## Introduction

Intimate partner violence (IPV) is a critical public health problem even in the 21st century, particularly in developing nations. This violation of human rights has been found to affect

cfm. Searching 'Bangladesh DHS, 2007' in the DHS website will provide the survey data set.

**Funding:** The author(s) received no specific funding for this work.

**Competing interests:** The authors have declared that no competing interests exist.

physical, mental and reproductive health of women. The World Health Organization (WHO) defines IPV as "any behavior within an intimate relationship that causes physical, psychological or sexual harm to those in the relationship" [1]. These behaviors include acts of physical violence, sexual violence, emotional or psychological abuse and controlling behaviors such as secluding the victim from family and friends, and/or restricting access to financial, educational, healthcare and personal freedom [1]. IPV is also known as domestic violence, wife abuse or dating violence [2]. However, in the current study we have considered only the physical aspects and sexual violence grounds as IPV for a more focused study (instead of the broad definition of the WHO), which is also more relevant in the context of Bangladesh [3]. It is postulated that IPV leads to an adverse effect on maternal and reproductive health, particularly miscarriages, induced abortions and/or stillbirths.

According to the WHO's global and regional estimates of violence against women, nearly 30% women around the globe have fallen victims to IPV, in forms of physical, sexual or emotional abuses, during the course of their relationships. In some regions, such as the South-East Asian and Eastern Mediterranean region, the prevalence of IPV increased by 38% [4]. The implications of IPV have been observed to exceed the apparent cases of physical injuries and mental health ailments, and have shown signs of affecting reproductive health issues. For example, women who have experienced IPV are more than twice as likely to have an induced abortion compared to women who have not been a victim [5].

IPV has resulted in a long list of health-related issues for women. These include increased risk of miscarriages, induced abortions, or stillbirths (MSA) [6], high blood pressure or edema, vaginal infection, severe nausea, kidney infection or urinary tract infection, preterm delivery and giving birth to a low-birthweight (LBW) infant [7], perinatal deaths, and preterm LBW deliveries [8]. Furthermore, psychological IPV were observed to have increased the odds of breastfeeding avoidance in Spain [9]. Female IPV victims were also exposed to birth control sabotage, forced sex, and partner's unwillingness to use birth control methods as well as at risk for HIV infection [10, 11].

The overall picture regarding women facing IPV in South-East Asia is bleaker compared to the developed countries. The regional prevalence rates IPV, by WHO region 2010, was the highest in South-East Asia at 37.7%, whereas the rate in the high-income countries was 23.2%. The South-East Asian region, primarily Bangladesh, India, Myanmar, Sri Lanka, Thailand, and Timor-Leste, was also found to have the highest median prevalence of intimate partner homicide among all female fatalities totaling to nearly 55% [5]. Women in Bangladesh who experienced sexual IPV during pregnancy were also at an increased risk of suffering from medical, obstetric or multifaceted complications during pregnancy [12]. However, there is a literature gap in identifying the most vulnerable cohorts of women who suffer from IPV in Bangladesh and its subsequent effect on miscarriages, induced abortions and stillbirths.

The status quo in Bangladesh regarding miscarriages, induced abortion and stillbirth (MSA) or fetal deaths continues to be challenging despite the focus in maternal healthcare from the policy makers. During 2014, an estimated number of induced abortions was 1,194,000 in Bangladesh, that is approximately 29 abortions per 1,000 women aged 15–49 years [13]. A combined analysis using Bangladesh Demographic and Health Survey (BDHS) 2004 to 2014 data found that the pooled rate of stillbirth in Bangladesh was 28 per 1000 births (95% CI: 22, 34) [14]. Although no substantial national statistics regarding miscarriage in Bangladesh were available, the rate of prevalence is reasonably assumed to be higher compared to developed countries. All these show the need for a better understanding of vulnerable cohorts for evident-based intervention strategies and refining maternal health policies in Bangladesh.

Prevalence of physical and/or sexual abuse during pregnancy is quite similar in industrialized and non-industrialized nations [15]. One study found that in rural Bangladesh, women's

social status and economic conditions could influence their risk of domestic violence in many ways [16]. A study in Bangladesh found that sociodemographic covariates such as age, place of residence, education, religion and number of children were associated with intimate partner violence [17].

The current study explored the vulnerable cohorts, that is, the households where women are vulnerable to IPV and consequently are likely to experience miscarriages, stillbirths or induced abortions (MSA) based on their sociodemographic attributes using Bangladesh Demographic and Health Survey 2007 (BDHS 2007). This should contribute to the Sustainable Development Goals (SDG), particularly goal 3.1 (reducing the global maternal mortality ratio) and goal 5.2 (eliminating all forms of violence against all women and girls in the public and private spheres) [18] by helping to design intervention strategies. Specifically, this study would provide (a) estimates of overall physical and sexual IPV victimization among ever-married Bangladeshi women in different socio-demographic scenarios and (b) an estimate of the effect of IPV on miscarriages, induced abortions and stillbirths (MSA) based on this victimization.

## Theoretical framework

Although using physical or sexual violence against wives is no longer legally permissible, the legacy of the patriarchy continues to generate the conditions and relationships that lead to IPV. Intimate partner violence, or wife abuse, was not publicly recognized as a social problem until the 1970s [19]. To understand the cause and mechanism of IPV, multiple theories, such as the systems theory [20], exchange theory [19], nested ecological theory [21], subculture-of-violence theory [22], and resource theory [23] are discussed by the sociologists within the paradigm of familial violence [24].

This study followed the nested ecological theory given by Dutton (2006). In his work, Dutton has identified the individual as the unit of analysis to discuss IPV and considered the environmental and societal factors to be significant in explaining the violent behavior in an intimate relationship. Dutton indicated four levels of systemic social framework that could affect the unit of analysis, the individual. They are: (a) The macrosystem, which is made up of "broad cultural values and belief systems"; (b) The exosystem, which consists of the institutions (educational, employment, religious) and groups (peers, society) through which the individual is connected to the outside environment; (c) The microsystem, which refers to the individual itself and interaction patterns within the family itself; and, (d) The ontogenetic factors, which refer to the "individual's developmental experience that shape responses to microsystem and exosystem stressors".

Typically, the selection of covariates in such studies of violence is based on applied knowledge or experience and literature review. However, in the current study, probable covariates were initially listed that could be associated with the outcome variables (IPV and MSA) based on the literature. Using the definitions of the systemic levels of social framework given in Dutton's nested ecological theory, the final list of sociodemographic variables for the study were made. The hierarchical tree shows the distribution of sociodemographic factors of this study among the four systemic levels of social framework (**Fig 1**).

## Methods and materials

### Ethical approval

This article does not contain any studies with human participants performed by any of the authors. The Bangladesh demographic and health Surveys were approved by ICF Macro Institutional Review Board and the National Research Ethics Committee of the Bangladesh Medical Research Council. A written consent about the survey was given by participants before the

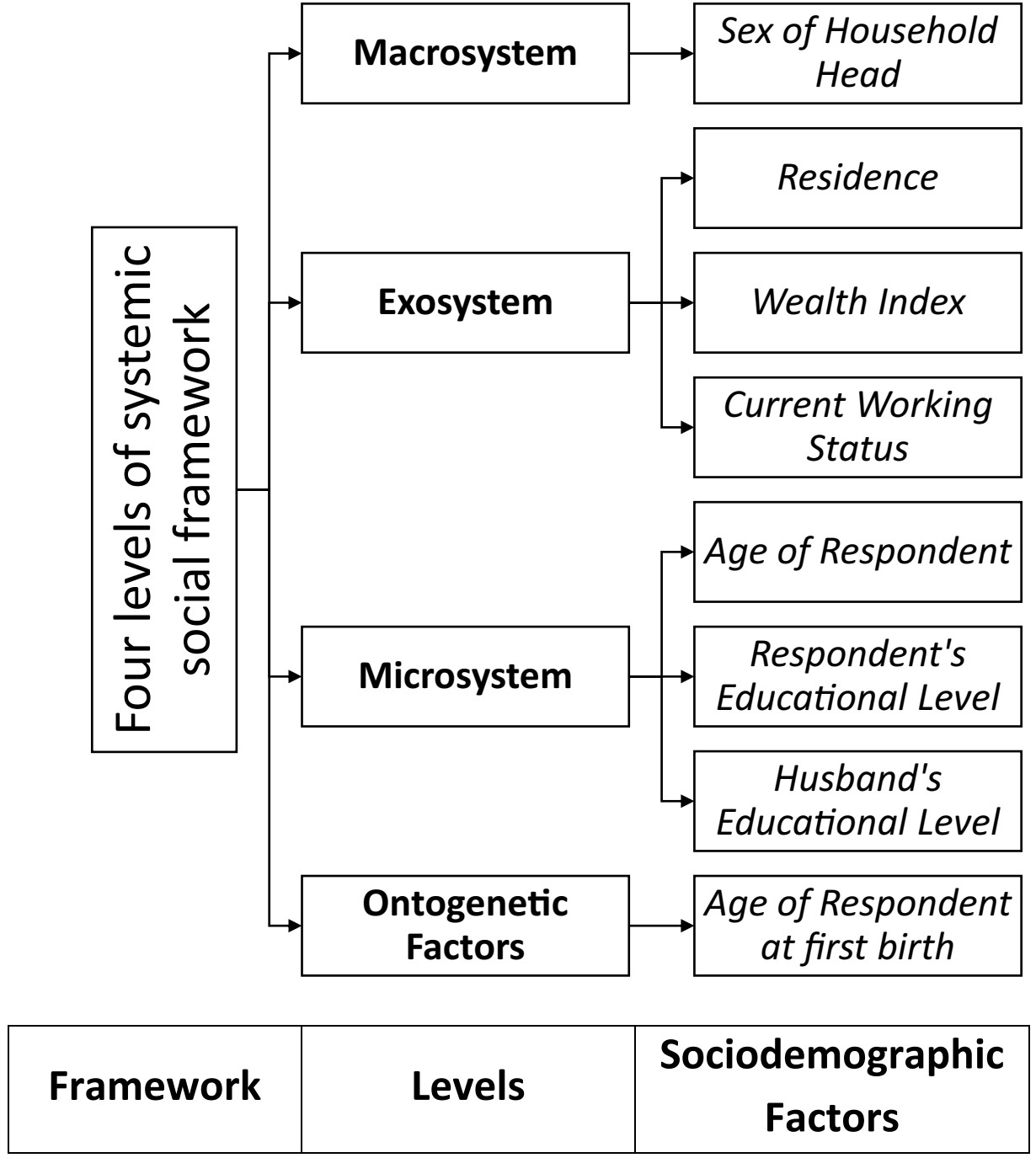

**Fig 1. Theoretical framework of the based on four levels of systemic social framework by Dutton (2006).**

interview. All identification of the respondents were dis-identified before publishing the data. The secondary data sets analysed during the current study are freely available upon request from the DHS website at http://dhsprogram.com/data/available-datasets.com. Searching 'Bangladesh DHS, 2007' in the DHS website will provide the survey data set.

## Data overview

The present study uses data from the fifth Bangladesh Demographic and Health Survey (BDHS) conducted in 2007 under the authority of the National Institute for Population Research and Training (NIPORT) in collaboration with the Demographic and Health Survey (DHS). This survey is a nationally representative household survey that has been periodically held since 1993 and uses a structured questionnaire. A list of enumeration areas (EAs) from the population census of Bangladesh conducted in 2001 was used as the sampling frame for the survey, where the whole country was divided into 6 administrative divisions and the sampling frame was comprised of 259,532 enumeration areas (EAs) with an average size of 100 households in each EA [25]. The survey used two-stage stratified cluster sampling, where 361 EAs (clusters) were selected in the first stage using the probability proportional to size method. Then in the second stage, an equal probability systematic sampling method was used to draw an average of 30 households from each of the 361 EAs.

All ever-married women aged 15–49 years who stayed in the selected household the night before the survey were eligible for the female survey. Interviews were successfully completed in 10,400 households (99.4 percent of the total selected households). In these households, a total of 11,178 women aged 15–49 were identified as eligible and finally 10,996 women were interviewed (a response rate of 98.4 percent). From them, 4,489 women were deemed eligible to respond to the domestic violence module where 22 women were excluded [25].

The study considered this sample of 4,489 women who responded to the questions regarding IPV and MSA. Women who were temporary (de jure) residents at the time of the survey were excluded from the current study. After removing respondents with missing values, the final sample for the study was 3,920 women.

## Response and outcome variables

Both IPV and MSA are outcome variables in this study. The outcome variable MSA was extracted from the original dataset while IPV was constructed from the available data. In order to group the partner's violent acts as IPV, the BDHS 2007 report as well as the DHS VII standard recode manual were followed [26]. The binary outcome variable IPV was constructed based on the respondents' answers to the following questions: (a) spouse ever pushed, shook or threw something; (b) spouse ever slapped; (c) spouse ever punched with fist or something harmful; (d) spouse ever kicked or dragged; (e) spouse ever tried to choke or burn; (f) spouse ever threatened with knife/gun or other weapon; (g) spouse ever physically forced sex when not wanted; and, (h) spouse ever twisted her arm or pulled her hair. An affirmative answer to any one of these questions was taken as 'yes' for IPV and the rest were classified as 'no'. Similarly, if the responded reported ever having a miscarriage or induced abortion or stillbirth, they were classified as 'yes' for the variable MSA and the rest were taken as 'no' (**Fig 2**).

The sociodemographic factors were selected based on prior literature, pre-analysis and the theoretical framework. The selected sociodemographic factors were respondent's current age (in years); place of residence (urban, rural); respondent's and her partner's highest educational level (no education, primary, secondary, higher); sex of household head (male, female); wealth index (poorest, poorer, middle, richer, richest); respondent's age at first birth (in years); and respondent's current working status (no, yes). The wealth index was pre-calculated in the data using principal component analysis on household assets [27].

## Statistical analyses

Bivariate analyses were conducted to observe the distribution of each covariate over the outcomes. The chi-square test and Cramer's V were used to assess the significance of associations between the sociodemographic variables and both outcome variables (IPV and MSA).

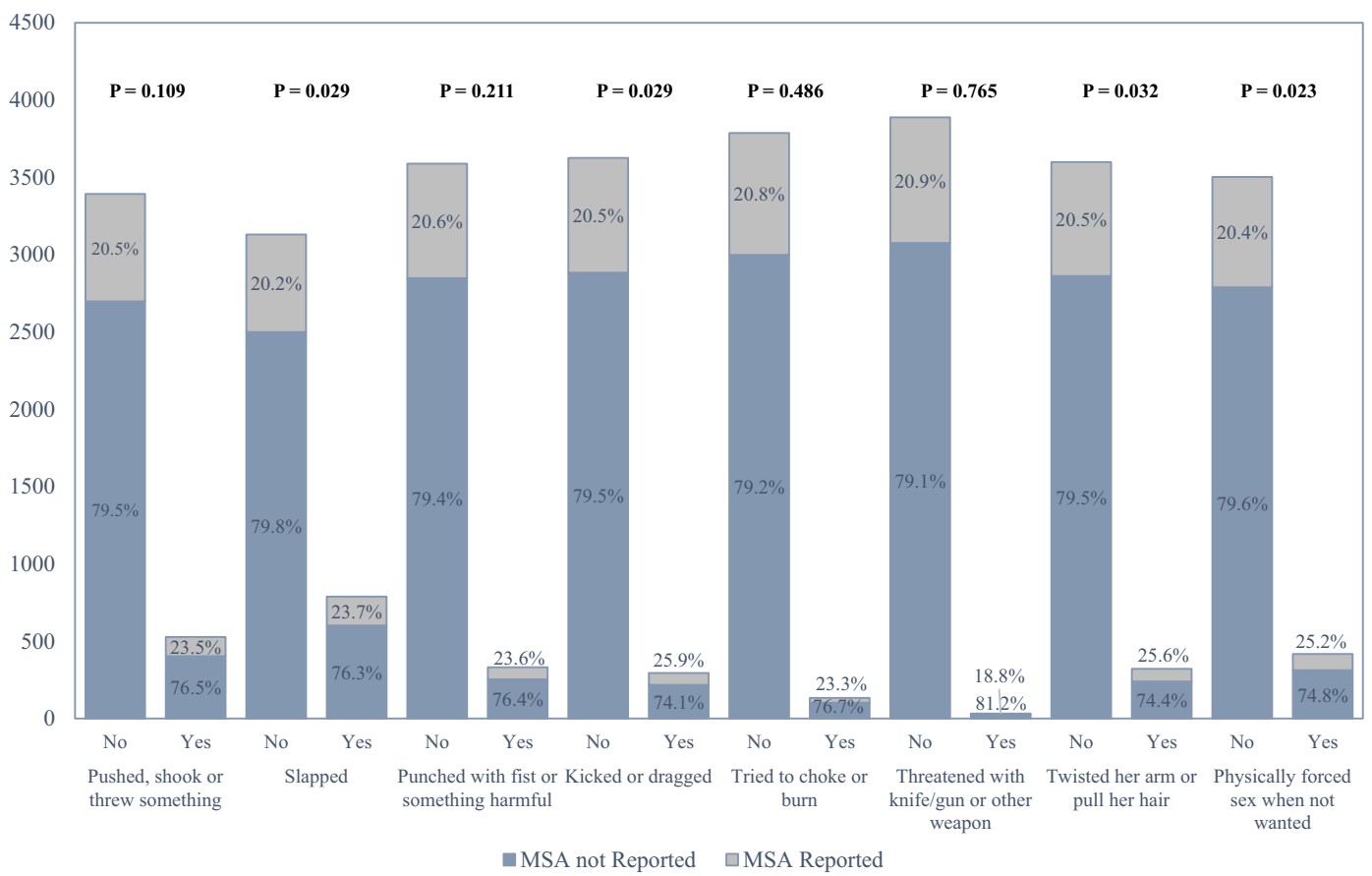

**Fig 2. Distribution of IPV among MSA in Bangladesh Demographic and Health Survey (BDHS) 2007.**

Both outcome variables in the study were binary, hence a regression model with a binomial family of distribution seemed appropriate. Generalized linear models (GLMs) were fitted to each outcome variable where cluster and strata-wise variations and survey weights were adjusted to generalize the findings in both models. For the first model with IPV as the outcome, the model accuracy was 73.6% and for the second model with MSA as the outcome, the model accuracy was 79.1%. GLMs are commonly used to fit multivariate distributions for non-normal data where random effects are incorporated into the linear predictors [28], and commonly used in DHS studies [29]. All analyses were conducted in *R* (*version 3.6.0*) and GLM was fitted using the "*survey*" package.

The present study employed a p-value threshold of 0.005, which is recommended for new discoveries in order to ensure reproducibility in scientific research [30]. Hence, associations were considered significant if p-values were less than or equal to 0.005 in the fitted models.

## Results

### Descriptive analysis

The prevalence of intimate partner violence was 26.4% (**Table 1**). The average age of respondents who were victims of IPV and reported at least one MSA during their lifetime preceding the survey was 31.6 years. The mean age at first birth was 17.8 years. More than half of the respondents (62.8%) lived in the rural areas of the country. Approximately one-third of the

**Table 1. Distribution of Intimate Partner Violence (IPV) victims and women with miscarriages, stillbirths and abortions (MSA) by sociodemographic factors (categorical and continuous), among women aged 15–49 years from the data obtained from BDHS 2007.**

| Sociodemographic variables | Group | N (%) of all study participants | N (%) of women subject to Intimate Partner Violence (IPV) | | N (%) of women with Miscarriages, Stillbirths and Abortions (MSA) | |
|---|---|---|---|---|---|---|
| | | | Yes | No | Yes | No |
| **Residence** | Urban | 1457 (37.2) | 353 (24.2) | 1104 (75.8) | 277 (19.0) | 1180 (81.0) |
| | Rural | 2463 (62.8) | 682 (27.7) | 1781 (72.3) | 542 (22.0) | 1921 (78.0) |
| | Cramer's V | | 0.04 | | 0.03 | |
| | p-value | | 0.017 | | 0.026 | |
| **Age of respondents (mean [SD])** | | 31.6 (8.6) | | | | |
| **Education** | No Education | 1400 (35.7) | 424 (30.3) | 976 (69.7) | 316 (22.6) | 1084 (77.4) |
| | Primary | 1204 (30.7) | 317 (26.3) | 887 (73.7) | 294 (24.4) | 910 (75.6) |
| | Secondary | 1060 (27.0) | 255 (24.1) | 805 (75.9) | 174 (16.4) | 886 (83.6) |
| | Higher | 256 (6.6) | 39 (15.2) | 217 (84.8) | 35 (13.7) | 221 (86.3) |
| | Cramer's V | | 0.09 | | 0.09 | |
| | p-value | | <0.001 | | <0.001 | |
| **Age of respondents at first birth (mean [SD)]** | | 17.8 (3.1) | | | | |
| **Sex of Household Head** | Male | 3490 (89.0) | 899 (25.8) | 2591 (74.2) | 726 (20.8) | 2764 (79.2) |
| | Female | 430 (11.0) | 136 (31.6) | 294 (68.4) | 93 (21.6) | 337 (78.4) |
| | Cramer's V | | 0.04 | | 0.01 | |
| | p-value | | 0.009 | | 0.691 | |
| **Wealth Index** | Poorest | 739 (18.9) | 270 (36.5) | 469 (63.5) | 180 (24.4) | 559 (75.6) |
| | Poorer | 772 (19.7) | 242 (31.3) | 530 (68.7) | 160 (20.7) | 612 (79.3) |
| | Middle | 732 (18.7) | 201 (27.5) | 531 (72.5) | 165 (22.5) | 567 (77.5) |
| | Richer | 740 (18.9) | 182 (24.6) | 558 (75.4) | 149 (20.1) | 591 (79.9) |
| | Richest | 937 (23.9) | 140 (14.9) | 797 (85.1) | 165 (17.6) | 772 (82.4) |
| | Cramer's V | | 0.17 | | 0.06 | |
| | p-value | | <0.001 | | 0.012 | |
| **Husband's Education** | No Education | 1400 (35.7) | 444 (31.7) | 956 (68.3) | 314 (22.4) | 1086 (77.6) |
| | Primary | 1072 (27.4) | 297 (27.7) | 775 (72.3) | 214 (20.0) | 858 (80.0) |
| | Secondary | 949 (24.2) | 222 (23.4) | 727 (76.6) | 206 (21.7) | 743 (78.3) |
| | Higher | 499 (12.7) | 72 (14.4) | 427 (85.6) | 85 (17.0) | 414 (83.0) |
| | Cramer's V | | 0.13 | | 0.04 | |
| | p-value | | <0.001 | | 0.059 | |
| **Current Work Status of Respondent** | Not Working | 2665 (68.0) | 646 (24.2) | 2019 (75.8) | 567 (21.4) | 2098 (78.6) |
| | Working | 1255 (32.0) | 389 (31.0) | 866 (69.0) | 252 (20.1) | 1003 (79.9) |
| | Cramer's V | | 0.07 | | 0.01 | |
| | p-value | | <0.001 | | 0.390 | |
| **Intimate Partner Violence (IPV)** | No | 2885 (73.6) | | | 569 (19.7) | 2316 (80.3) |
| | Yes | 1035 (26.4) | | | 250 (24.2) | 785 (75.8) |
| | Cramer's V | | | | 0.05 | |
| | p-value | | | | 0.003 | |
| **Total sample size** | 3920 | | | | | |

study females lacked formal education, and the rest had some form of education (primary, secondary or higher), with only 6.7% having higher level education. In only 11% of the cases, the household head was a female member of the family.

The distribution of partner's education status was similar to that of the respondents, as nearly one-third of the partners reported having no formal education. However, in this case

the partners' highest level of education (higher 12.7%) was almost double than that of the respondents. Even though about two-thirds of the respondents had some sort of formal education, only 32% respondents reported that they were currently working outside of homes in some capacity.

The lifetime prevalence of IPV among these various sociodemographic groups presents an interesting story. Around 24% of study participants who resided in urban areas reported facing some form of IPV during their lifetime, whereas it was higher (27.7%) in the rural areas. Women with higher levels of education faced fewer IPV (15.2%) compared to those who were illiterate (30.3%). IPV was higher (31.6%) in households where a female member was the head of the household than a household with a male house head (25.8%). About 36.5% women from the poorest wealth group reported IPV, more than 10% higher than the overall prevalence. These figures gradually decreased in the subsequent wealthier households and were reported to be the lowest in the richest group (14.9%). Respondents who had partners with higher education reported the least prevalence of IPV (14.4%) compared to the preceding lower education groups where it was as high as 31.7% (no education). Women who reported working outside of home faced a higher prevalence of IPV (31%) compared to women who were not working (24.2%) at the time of the survey.

The overall prevalence of at least one or more of either miscarriages, stillbirths and/or induced abortions (MSA) among the study participants was approximately 21%. The prevalence of MSA among urban (19%) and rural (22%) residents was not considerably different from the overall prevalence. However, the rate declined sharply from primary education group to secondary and higher education groups. Here, the primary education group reported a prevalence of MSA at 24.4%, while the higher education group reported the lowest prevalence at 13.9%. With regards to the wealth index, MSA was the highest for the poorest group (24.4%) and the lowest for the richest group (17.6%), while the three intermediary groups reported figures close to the overall rate (~21%). Women who were subject to IPV reported an MSA prevalence rate of 24.2%, while women who were not a victim of IPV had an MSA prevalence rate of 19.7%.

## The generalized linear model

Respondent's current age, residence, sex of household head and wealth index were found to be significantly associated with IPV in the GLM (**Table 2**). However, respondent's current age, respondent's age at first birth and intimate partner violence were significantly associated with MSA. Younger females were more likely to be victims of IPV than older females (OR 0.96 with 95% CI: 0.95, 0.97). Women residing in urban areas were 28% less likely to face IPV than rural women (p-value = 0.004). The likelihood of IPV also significantly reduced for women who belonged to the richer (OR 0.68) and richest (OR 0.31) wealth groups compared to women from the poorest households. In households where a female member was considered the head, the odds of IPV significantly increased by 1.56 times compared to households with male heads. Older women were at increased risk of MSA, as were women who gave birth at an older age, which is understandable since the risk of MSA increases as age increases. The odds of having a miscarriage, stillbirth and/or induced abortion significantly increased by 35% (p-value = 0.003) for women who were victims to IPV compared to women who faced no such violence.

## Discussion

In order to sustain the progress in public health paradigm and improve maternal healthcare, and pave the way to meet the goal 3.1 of reducing the global maternal mortality ratio and goal 5.2 of eliminating all forms of violence against women in both public and private spheres of

**Table 2. Generalized Linear Model (GLM) fitted with Intimate Partner Violence (IPV) and miscarriages, stillbirths and abortions (MSA) to sociodemographic factors where cluster and strata-wise variations and survey weights were adjusted.**

| Sociodemographic factors | Intimate Partner Violence (IPV) | | Miscarriages, Abortion, Stillbirth | |
|---|---|---|---|---|
| | Odds Ratio (95% CI) | p-value | Odds Ratio (95% CI) | p-value |
| **Age** | 0.96 (0.95, 0.97) | <0.001 | 1.03 (1.02, 1.04) | <0.001 |
| **Residence (ref: Rural)** | | | | |
| Urban | 0.72 (0.57, 0.90) | 0.004 | 1.28 (1.01, 1.62) | 0.047 |
| **Education (ref: No Education)** | | | | |
| Primary | 0.88 (0.70, 1.10) | 0.266 | 1.20 (0.96, 1.51) | 0.111 |
| Secondary | 0.98 (0.74, 1.29) | 0.873 | 0.79 (0.57, 1.09) | 0.148 |
| Higher | 1.15 (0.69, 1.92) | 0.594 | 0.52 (0.27, 0.97) | 0.041 |
| **Sex of Household Head (ref: Male)** | | | | |
| Female | 1.56 (1.19, 2.05) | 0.002 | 0.99 (0.71, 1.37) | 0.936 |
| **Wealth Index (ref: Poorest)** | | | | |
| Poorer | 0.85 (0.66, 1.09) | 0.203 | 0.77 (0.55, 1.09) | 0.144 |
| Middle | 0.76 (0.59, 0.98) | 0.033 | 0.91 (0.67, 1.22) | 0.531 |
| Richer | 0.68 (0.52, 0.88) | 0.004 | 0.80 (0.56, 1.13) | 0.210 |
| Richest | 0.31 (0.22, 0.43) | <0.001 | 0.80 (0.55, 1.16) | 0.238 |
| **Age of respondent at first birth** | 0.96 (0.92, 0.99) | 0.019 | 1.05 (1.02, 1.08) | 0.002 |
| **Husband's Educational Level (ref: No Education)** | | | | |
| Primary | 0.78 (0.64, 0.94) | 0.012 | 0.90 (0.69, 1.17) | 0.432 |
| Secondary | 0.83 (0.65, 1.07) | 0.152 | 1.13 (0.87, 1.48) | 0.353 |
| Higher | 0.63 (0.41, 0.97) | 0.037 | 1.14 (0.72, 1.81) | 0.576 |
| **Current Work Status of Respondent (ref: Not Working)** | | | | |
| Working | 1.17 (0.96, 1.42) | 0.117 | 0.91 (0.75, 1.12) | 0.392 |
| **Intimate Partner Violence (ref: No)** | | | | |
| Yes | | | 1.35 (1.11, 1.64) | 0.003 |

the Sustainable Development Goals (SDG) [18], there is a urgency to identify the vulnerable cohorts and reinforce interventions programs by focusing on the specific needs of each cohort in Bangladesh. With the objective of identifying the cohort of vulnerable women in terms of IPV and miscarriages, stillbirths and induced abortion, this study observed significant association between these factors and age of women, residence, age of women at their first birth, sex of household head and household's financial condition. Interestingly, MSA was also significantly associated with IPV, which indicates that a combined intervention effort might be comprehensive in the targeted cohorts.

According to the findings, respondent's age was significantly associated with IPV as younger women seem to be more vulnerable to IPV than older women. A recent study with nine countries of the WHO multi-country study also concluded that adolescent women are at a greater risk of facing IPV than older women [31]. Similarly, respondent's current age and age at first birth were also found to be significant factors for miscarriages, stillbirths and induced abortion (MSA). There is a myriad of literature that estimates that increase in age of women makes them prone to miscarriages and induced abortion [32–35].

Moreover, many literature also conclude that an increase in maternal age raises the risk of stillbirths or fetal deaths [36, 37]. Since child marriage for females is still widespread in Bangladesh with mean age at first marriage at 15.7 years [38], young women are subject to miscarriages, stillbirths and induced abortions. This could also explain the prevalence of IPV of young women; as they get married off early, they are unlikely to have maturity or financial stability to confront the patriarchal environment in their in-laws households [39].

The place of residence was observed as another considerable factor for IPV in Bangladesh; however, it was not significant in the case of MSA. The odds of urban women experiencing IPV was substantially lower in comparison to rural women, which is consistent with previous literature [6, 40]. This could be explained by the fact that women in urban areas are significantly less compliant to domestic violence due to their increased education, decision making autonomy, and partner's higher education, as justification of spousal violence were found significantly linked to these factors [39, 41]. It could be further postulated that female rights organizations and media have a stronger presence in the metropolitan compared to distant villages where IPV is often considered as a part of traditional norm [42, 43].

Results revealed that women living in households with female heads were more vulnerable to IPV than households with male heads. However, some studies had observed the opposite scenario that high prevalence of child marriage and poverty lead to wife beating in patriarchal households [44, 45]. A recent study regarding the patterns of IPV in Bangladesh suggests that even though there is an increase in the awareness among women regarding their rights, women still continue to face social stigma and barriers that confine them from exercising these rights [46].

It could be hypothesized that patriarchal household norms still dictate the social behavior towards women, and households with female heads either struggle to ensure a safe household for women due to these archaic social norms and beliefs, or it could be that the male partners exert their frustration of living in a household led by a female member in a form of IPV on their female counterparts in these households. Moreover, female house heads are less likely to be highly educated compared to male house heads, which might bar them from going against the tradition. Generally, men in patriarchal societies resort to the idea of masculinity that involves specified hierarchical gender positions and measuring male success in terms of the power of exerting control over women [47]. These pose an interesting issue from the perspective of women empowerment and policy implementation emphasizing on eliminating such beliefs, which would require participation of men besides women to decrease the risk of IPV in Bangladesh [48].

In the context of household wealth, women in richer quintiles reported significantly reduced IPV compared to women from poorer households. It could be argued that households with higher income or wealth engage in less resource related disputes and this leads to decreased IPV as studies on Bangladeshi women corroborate that household financial condition acts as a protection against IPV [44]. Poverty in Bangladesh is entwined with dowry as poor bridegrooms expect hefty monetary payment from the girl's father and often a delay or lack of payment leads to IPV, particularly in rural areas [49]. Policies aiming at eradicating traditional norms in remote areas as well as poverty reduction are to be considered to tackle these issues.

Higher IPV was found to significantly associated with greater the odds of reporting one or more of miscarriage, stillbirths and induced abortions (MSA), which was a primary hypothesis in the present study. Globally, women are subject to IPV during pregnancy and it is considered one of the factors for negative maternal health outcomes [50, 51]. There have been multiple studies that reported increased adverse obstetric outcomes for women who were subject to IPV during pregnancy [52–54]. Furthermore, it was suggested that termination of pregnancy was significantly associated with IPV for women in Bangladesh [55].

The significant association between IPV and MSA corroborates with the previous findings regarding the relation between reproductive health choices of a woman and outcomes that can lead to MSA. One probable reasoning could be that abusive relationships could lead to scenarios where women cannot make decisions regarding their sexual lives and reproductive health and would then result in unplanned or unwanted pregnancies, ultimately leading to MSA [6].

Moreover, in relationships where females are subjected to IPV, the pregnancy could be in jeopardy as a result of physical and sexual IPV. Furthermore, in a number of cases, the partners could force women to terminate their pregnancies as women in abusive relationships may not have the power to confront the excessive pressures from their partners. In the case of Bangladesh, such scenarios are not uncommon and women are vulnerable to these adverse effects, since women's lack of autonomy in decision making and accepting mentality to patriarchal family dynamics are still prevalent [56, 39].

The study had a few limitations. Firstly, since BDHS 2007 is a cross sectional survey, a causal relationship between IPV and MSA could not be estimated. Secondly, the timing and chronology of IPV and reported MSA also could not be established due to lack of data. Therefore, the direction of IPV and MSA relationship could not be determined which would provide a clearer understanding. Hence, we need to exercise caution during interpretation. Thirdly, BDHS 2007 was the latest data set that collected data on IPV in Bangladesh, which demands recent DHS to include such information. Finally, there is the case of stigmatization in reporting both IPV and MSA among Bangladeshi women, where some might refrain from providing information on these sensitive issues [6]. Hence, the cases of IPV or MSA could be higher than reported in BDHS 2007. Furthermore, there are restrictions on induced abortions in Bangladesh by law and so these numbers could be underreported [55]. Future studies could venture the possible mental health aspects of IPV and collect data to extend this study.

## Conclusion

Bangladesh has made great strides in reducing child and maternal mortality in the 21st century; however, more needs to be done to reach the standards globally set to attain SGDs. The objective of this study was to identify the most vulnerable cohorts in danger of IPV and MSA by studying women's pertinent sociodemographic aspects and understanding the effects of IPV on maternal health and adverse obstetric outcomes. Keeping in mind that these data are subject to being underreported and the actual numbers could be relatively higher, the findings demand that, in order to protect women from IPV, dedicated interventions and targeted policy frameworks are required for young women from rural areas, particularly from poorer and underprivileged households.

Awareness and social education campaigns have shown to work against traditional norms regarding the position of women in households, women empowerment, and early marriages. As IPV and MSA are intertwined with each other, policies aiming at intervening the vulnerable cohorts for IPV and improving maternal health are likely to improve the overall maternal health situation and could contribute in achieving SDG goals 3.1 and 5.2 set by the United Nations [18].

It would be beneficial for future studies if DHS continued collecting data on domestic violence against women, so that further studies could be done to observe the trend of IPV in Bangladesh and conduct appropriate comparisons over the years for evidence based policy making. The study also encourages more focused work on the chronology of IPV and MSA in order to get a better understanding on the strength and direction of the association between them and test possible intervention strategies in the context of Bangladesh.

## Acknowledgments

The authors would like to acknowledge MEASURE Evaluation, National Institute for Population Research and Training (NIPORT) and USAID/Bangladesh, who allowed the researchers to access the survey data for free.

## Author Contributions

**Conceptualization:** Awan Afiaz, Raaj Kishore Biswas.

**Data curation:** Awan Afiaz, Raaj Kishore Biswas.

**Formal analysis:** Awan Afiaz.

**Investigation:** Nurjahan Ananna.

**Methodology:** Awan Afiaz.

**Software:** Awan Afiaz, Raaj Kishore Biswas.

**Supervision:** Raaj Kishore Biswas.

**Visualization:** Awan Afiaz.

**Writing – original draft:** Awan Afiaz, Raaj Kishore Biswas.

**Writing – review & editing:** Awan Afiaz, Raaj Kishore Biswas, Raisa Shamma, Nurjahan Ananna.

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
