## [Decision Letter · Decision Letter 0]

21 May 2020

PONE-D-20-07575

Intimate partner violence (IPV) with miscarriages, stillbirths and abortions: Identifying vulnerable households for women in Bangladesh

PLOS ONE

Dear Dr. Awan,

Thank you for submitting your manuscript to PLOS ONE. After careful consideration, we feel that it has merit but does not fully meet PLOS ONE’s publication criteria as it currently stands. Therefore, we invite you to submit a revised version of the manuscript that addresses the points raised during the review process.

We look forward to receiving your revised manuscript.

Kind regards,

Russell Kabir, PhD

Academic Editor

PLOS ONE

2. In the ethics statement in the manuscript and in the online submission form, please confirm whether all data were fully anonymized BEFORE you accessed them.

3. Please provide a sample size and power calculation in the Methods, or discuss the reasons for not performing one before study initiation.

Reviewers' comments:

Reviewer's Responses to Questions

**Comments to the Author**

1. Is the manuscript technically sound, and do the data support the conclusions?

Reviewer #1: Yes

Reviewer #2: No

2. Has the statistical analysis been performed appropriately and rigorously? 

Reviewer #1: Yes

Reviewer #2: I Don't Know

3. Have the authors made all data underlying the findings in their manuscript fully available?

Reviewer #1: No

Reviewer #2: Yes

4. Is the manuscript presented in an intelligible fashion and written in standard English?

Reviewer #1: Yes

Reviewer #2: No

5. Review Comments to the Author

Reviewer #1: The authors have addressed an important issue and this manuscript is a sound presentation. The abstract is well explained and concise.

However in figure 1, I would suggest boxes with white background and black text inside.

In figure 2, please add the p values if possible between the Yes and No column with asterisks.

There are some grammatical errors and I would suggest to go through again for during minor revision.

Reviewer #2: This topic is important in terms of developing regions as IPV is prevalent in these regions. After going through the article, I am wondering at some points.

Comments:

1. The introduction section is redundant in some aspect. However, the study did not add previous studies regarding the subject matter. What is the actual gap of the study is missing?

2. The theoretical framework is resourceful in terms of theories. How it linked to the present studies, the figure 1 does not provide enough information.

3. In methodology section, how outcome variables were calculated and measure need to explain. In statistical analysis section, model fit information need to add.

4. In Table 1, better to report the Phi coefficient and Cramer's V (Strength of association) along with the p-value obtained from the Chi-square test.

5. Repetition of some information in introduction, discussion and conclusion section might degrade the quality.

6. This study is based on the data of BDHS 2007. What this study adds in terms of current policy implication while Bangladesh is doing better over most of the selected indicators.

6. PLOS authors have the option to publish the peer review history of their article (what does this mean?). If published, this will include your full peer review and any attached files.

Reviewer #1: No

Reviewer #2: No

---

## [Author Response · Author response to Decision Letter 0]

12 Jun 2020

May 23, 2020

Russell Kabir, PhD

Academic Editor

PLOS ONE

RE: PLOS ONE Decision: Revision required [PONE-D-20-07575]

Dear Editor,

Thank you for providing us with the opportunity to revise our manuscript titled “Intimate Partner Violence (IPV) with miscarriages, stillbirths and abortions: Identifying vulnerable households for women in Bangladesh” for publication in your esteemed journal “PLOS ONE”. 

We have addressed the reviewers’ comments in our response and have incorporated the required changes into the manuscript through careful revisions as prescribed by both reviewers and provided a copy of the manuscript indicating track changes. For convenience, the reviewers’ comments are highlighted in blue and our response is written in black coloured font. 

The primary changes made in the revised version of the manuscript include the following: 

1. As per suggestions from Reviewer #1, we made necessary changes to both figures, 

2. Few sentences were added in Theoretical Framework and Method section for clarification as suggested by Reviewer #2,

3. Additional information was added to Table 1 according to the suggestions made by Reviewer #2.

In addition to the above changes, some minor changes throughout the revised version of the manuscript were made, mostly to correct a number of minor issues pointed out by the reviewers as well as fix some grammatical issues.

Please convey our most sincere gratitude to the reviewers for their valuable feedbacks and suggestions in order to improve the manuscript. 

We cordially thank you for your patience and kind consideration during these unprecedented times. Please stay safe.

Sincere regards, 

Awan Afiaz

Institute of Statistical Research and Training,

University of Dhaka, 

Neelkhet Road, 

Dhaka, 1000, Bangladesh.

Email: aafiaz@isrt.ac.bd

Co-authors:

1. Raaj Kishore Biswas, Transport and Road Safety (TARS) Research Centre, School of Aviation, University of New South Wales, Australia

2. Raisa Shamma, Institute of Statistical Research and Training, University of Dhaka, Bangladesh

3. Nurjahan Ananna, Ibrahim Medical College, Dhaka, Bangladesh

 

Intimate Partner Violence (IPV) with miscarriages, stillbirths and abortions: Identifying vulnerable households for women in Bangladesh

[PONE-D-20-07575]

Reviewer #1

The authors have addressed an important issue and this manuscript is a sound presentation. The abstract is well explained and concise.

However in figure 1, I would suggest boxes with white background and black text inside.

In figure 2, please add the p values if possible between the Yes and No column with asterisks.

There are some grammatical errors and I would suggest to go through again for during minor revision.

Reviewer’s response: 

Thank you for the valuable inputs. The issue discussed in the manuscript is indeed an important discussion regarding women’s health and safety in the context of Bangladesh. 

1. In Figure 1 we have used white background with black text inside as suggested. 

2. In Figure 2, we have added the respective p-values between the Yes and No columns. However, instead of asterisks we have put the values on top the joint bars to make them more visually appealing. 

In response to addressing grammatical issues, we have consulted an English language expert and made necessary changes to the manuscript. 

 

Reviewer #2

This topic is important in terms of developing regions as IPV is prevalent in these regions. After going through the article, I am wondering at some points.

Comments:

1. The introduction section is redundant in some aspect. However, the study did not add previous studies regarding the subject matter. What is the actual gap of the study is missing?

Reviewers’ response: 

Thank you pointing that. The purpose of the introduction section was to gradually build up to the main idea behind the present study, which was to identify the groups of women vulnerable to intimate partner violence (IPV) in Bangladesh and address the fact that there exists a lack of available literature regarding the aforementioned vulnerable groups and shed light on the fact that there lies an important association with IPV and women’s experience of miscarriages, stillbirths and abortions (MSA). 

The previous study pertinent to the subject matter of the current study was mentioned in the introduction section with one similar study noted in reference 6, “Silverman, J. G., Gupta, J., Decker, M. R., Kapur, N., & Raj, A. (2007). Intimate partner violence and unwanted pregnancy, miscarriage, induced abortion, and stillbirth among a national sample of Bangladeshi women.” 

However, this was the only such notable study that discussed the association between IPV and MSA, albeit with a different survey. No other studies explored the sociodemographic characteristics of women in Bangladesh who experienced IPV using a nationally representative survey. We duly indicated this fact in our manuscript. In page 4, line 77, of the revised manuscript with track changes, we declared this lack of availability of previous pertinent studies, stating, “However, there is a literature gap in identifying the most vulnerable cohorts of women who suffer from IPV in Bangladesh and its subsequent effect on miscarriages, induced abortions and stillbirths.”

Then in the final paragraph of the Introduction section, we discussed our study objective. In page 5, line 102 of the revised manuscript with track changes, we stated, “The current study explored the vulnerable cohorts, that is, the households where women are vulnerable to IPV and consequently are likely to experience miscarriages, stillbirths or induced abortions (MSA) based on their sociodemographic attributes using Bangladesh Demographic and Health Survey 2007 (BDHS 2007). The understanding of vulnerable groups would help design sustainable interventions and subsequent policies to decrease the prevalence of IPV and improve maternal healthcare.”

Our current analysis contributes to the existing gap in literature. In our final paragraph of the conclusion we mentioned the motivation behind our paper asking the DHS to continue collecting relevant data on IPV and MSA, since such data were not collected in the following three rounds of survey since 2007. Our aim is to raise awareness regarding this neglected issue and encourage further dedicated studies and intervention to properly address this problem.

2. The theoretical framework is resourceful in terms of theories. How it linked to the present studies, the figure 1 does not provide enough information.

Reviewers’ response: 

The main idea behind the inclusion of the framework was to provide a sound theoretical basis for the selection of the study variables. While most sociodemographic studies use pre-existing literature or experience while selecting the variables, this practice does not help in designing targeted research efforts or intervention programs due to the lack of theoretical basis. To clarify the motivation behind this section in the paper we added (line 137, page 7 of the revised manuscript with track changes) that we wanted to solidify the choice of variables not just from the literature or our past experience, rather from existing theory.

We noted this idea in the final paragraph (page 7, line 140 of the revised manuscript with track changes) of the theoretical section, stating, “Using the definitions of the systemic levels of social framework given in Dutton’s nested ecological theory, the final list of sociodemographic variables for the study were made.”

To link the present study with theories in the 70’s we used Figure 1. While the ‘framework’ and ‘levels’ come from Dutton’s nested ecological theory, the ‘sociodemographic factors’ are from the present study. We have shown the links how each level connects with each of the sociodemographic factor. Thank you for raising the point. We hope that after updating the figure as suggested by reviewer 1, we were clearer in making our case.

3. In methodology section, how outcome variables were calculated and measure need to explain. In statistical analysis section, model fit information need to add.

Reviewers’ response:

Thank you for bringing these issues to our attention. The outcome variable was constructed from the original dataset (IPV) or readily available (MSA) and was readily available for use after removing the case wise missing observations. To clarify this issue, we have added the following sentences in the ‘Response and Outcome Variables’ subsection (page 9, line 182-183 of the revised manuscript with track changes)

“Both IPV and MSA are outcome variables in this study. The outcome variable MSA was extracted from the original dataset while IPV was constructed from the available data.” 

 Please note that in line 183, page 9 of the revised manuscript with track changes, we showed how the binary outcome variable IPV was constructed: “The binary outcome variable IPV was constructed based on the respondents’ answers to the following questions: (a) spouse ever pushed, shook or threw something; (b) spouse ever slapped; (c) spouse ever punched with fist or something harmful; (d) spouse ever kicked or dragged; (e) spouse ever tried to choke or burn; (f) spouse ever threatened with knife/gun or other weapon; (g) spouse ever physically forced sex when not wanted; and, (h) spouse ever twisted her arm or pulled her hair.”

For model fit information we revised the sentence in page 10-11, line 210-213 of the revised manuscript with track changes, in the ‘Statistical Analyses’ subsection as follows: 

“Generalized linear models (GLMs) were fitted to each outcome variables where cluster and strata-wise variations and survey weights were adjusted to generalize the findings in both models. For the first model with IPV as the outcome, the model accuracy was 73.6% and for the second model with MSA as the outcome, the model accuracy was 79.1%.”

4. In Table 1, better to report the Phi coefficient and Cramer's V (Strength of association) along with the p-value obtained from the Chi-square test. 

Reviewers’ response:

Thank you for your suggestion. We have added the Cramer’s V statistic as well. 

Given we have reported the Chi-square significance, reporting the Phi coefficient would be redundant in this case. Furthermore, the interpretation of Phi coefficients is not straightforward. Thus, we are happy to comply with your suggestions to report Cramer’s V besides the initial results of the Chi-square significance. 

5. Repetition of some information in introduction, discussion and conclusion section might degrade the quality.

Reviewers’ response:

Thank you for pointing that. We have thoroughly revised Introduction and Discussion to address this issue. We would like to point out that some were added for better flow of the paper; however, we respect your suggestion and hence removed those which were repetitive.

Please note that in the revised manuscript with track changes, we have removed the paragraph in page 5, lines 93-98, in compliance with the above suggestion. Next, we removed another sentence in page 6, lines 108-110 as well. Furthermore, we removed a sentence in page 22, lines 380-382. 

6. This study is based on the data of BDHS 2007. What this study adds in terms of current policy implication while Bangladesh is doing better over most of the selected indicators.

Reviewers’ response:

Thank you for pointing that. We agree with you. To do better policy analysis, we need recent data. That is why we wanted to use our study findings to inform data custodians that they need to continue collecting data on IPV and MSA. Unfortunately, DHS authorities have not collected information on IPV and MSA following 2007 survey. Thus, we stated the following in line 391, page 22 of the revised manuscript with track changes,

“It would be beneficial for future studies if DHS continued collecting data on domestic violence against women, so that further studies could be done to observe the trend of IPV in Bangladesh and conduct appropriate comparisons over the years for evidence based policy making.”

We whole-heartedly thank Reviewer 1 and 2 for their contributions and suggestions for improving the manuscript.

---

## [Editor Report · Decision Letter 1]

13 Jul 2020

Intimate partner violence (IPV) with miscarriages, stillbirths and abortions: Identifying vulnerable households for women in Bangladesh

PONE-D-20-07575R1

Dear Dr. Awan,

We’re pleased to inform you that your manuscript has been judged scientifically suitable for publication and will be formally accepted for publication once it meets all outstanding technical requirements.

Kind regards,

Russell Kabir, PhD

Academic Editor

PLOS ONE
---

## [Editor Report · Acceptance letter]

15 Jul 2020

PONE-D-20-07575R1 

Intimate partner violence (IPV) with miscarriages, stillbirths and abortions: Identifying vulnerable households for women in Bangladesh 

Dear Dr. Afiaz:

I'm pleased to inform you that your manuscript has been deemed suitable for publication in PLOS ONE. Congratulations! Your manuscript is now with our production department. 

Kind regards, 

on behalf of

Dr. Russell Kabir 

Academic Editor

PLOS ONE